# Development and Evaluation of the Biological Activities of a Plain Mucoadhesive Hydrogel as a Potential Vehicle for Oral Mucosal Drug Delivery

**DOI:** 10.3390/gels10090574

**Published:** 2024-09-03

**Authors:** Ana G. Pardo-Rendón, Jorge L. Mejía-Méndez, Edgar R. López-Mena, Sergio A. Bernal-Chávez

**Affiliations:** 1Departamento de Ciencias Químico Biológicas, Universidad de las Américas Puebla, San Andrés Cholula 72810, Puebla, Mexico; ana.pardorn@udlap.mx (A.G.P.-R.); jorge.mejiamz@udlap.mx (J.L.M.-M.); 2Programa de Edafología, Colegio de Postgraduados, Campus Montecillo, Carr. México Texcoco km 36.4, Montecillo 56230, Mexico; 3Tecnológico de Monterrey, Escuela de Ingeniería y Ciencias, Av. Gral. Ramón Corona No. 2514, Colonia Nuevo México, Zapopan 45121, Jalisco, Mexico

**Keywords:** oral mucosa, hydrogels, cationic guar gum, factorial regression analysis, biological activities

## Abstract

This study aimed to develop HGs based on cationic guar gum (CGG), polyethylene glycol (PEG), propylene glycol (PG), and citric acid (CA) using a 2^k^ factorial experimental design to optimize their properties. HGs were characterized through FTIR and Raman spectroscopy, scanning electron microscopy (SEM), and thermogravimetric analysis (TGA). The biological activities of HGs were determined by evaluating their mucoadhesive capacity and antibacterial activity in vitro, whereas their toxicity was analyzed using *Artemia salina* nauplii as an in vivo model. Results revealed that HGs were successfully optimized for their viscosity, pH, and sensory properties, and it was observed that varying concentrations of PEG-75 did not influence them. Through SEM analyses, it was noted that increased levels of PEG-75 resulted in HGs with distinct porosity and textures, whereas FTIR and Raman spectroscopy exhibited representative peaks of the raw materials used during the synthesis process. TGA studies indicated the thermal stability of HGs, as they presented degradation patterns at 100 and 300 °C. The synthesized HGs exhibited similar mucoadhesion kinetic profiles, demonstrating a displacement factor at an equilibrium of 0.57 mm/mg at 5 min. The antibacterial activity of HGs was appraised as poor against Gram-positive and Gram-negative bacteria due to their MIC_90_ values (>500 μg/mL). Regarding *A. salina*, treatment with HGs neither decreased their viability nor induced morphological changes. The obtained results suggest the suitability of CGG/PEG HGs for oral mucosa drug delivery and expand the knowledge about their mucoadhesive capacity, antibacterial potential, and in vivo biocompatibility.

## 1. Introduction

Oral mucosa performs multiple vital functions for general health and well-being. Anatomically, the oral mucosa consists of the oral epithelium, lamina propria, and submucosa [1]. Physiologically, it acts as a physical and immunological barrier against a wide range of external stimuli, including mechanical, chemical, and biological agents, as well as carcinogenic substances and oral bacteria [2,3]. Additionally, the oral mucosa plays a key role in secretion, housing salivary glands responsible for producing saliva, which maintains tissue moisture, facilitates digestion and lubrication, and plays an immunological role [4].

Saliva is a complex secretion that protects soft tissues, controls dryness, and can influence tissue repair. Composed of 99% water and 1% organic and inorganic molecules, saliva has a pH range between 7 and 7.4 [5]. A persistent decrease in pH can lead to symptoms such as cervical caries, gingival recession, cervical erosion, demineralization, and white spots on enamel [6]. In contrast, mucin, a complex and viscous substance produced and secreted by specialized cells in columnar epithelia, plays a crucial role in protecting underlying epithelial tissues by providing lubrication, hydration, and a barrier against harmful agents and pathogens [7].

Biochemically, mucin is abundant in serine, threonine, and proline, which are aminoacidic residues extensively glycosylated with fucose, galactose, sialic acids, *N*-acetylglucosamine (GlcNAc), and *N*-acetylgalactosamine (GalNAc) [8,9]. Sulfate ester groups and carboxyl groups of sialic acids in oligosaccharides confer a net negative charge to mucin, varying with oligosaccharide composition [10,11]. This negative charge is crucial for drug formulation in oral mucosa, as it necessitates positively charged carriers for effective adhesion and drug delivery effectiveness [12]. For example, recent reports have documented that positively charged systems such as chitosan nanoparticles, polymethacrylate micelles, and β-cyclodextrin/dialdehyde interact electrostatically with negatively charged mucin, enhancing the retention and bioavailability of therapeutic agents such as bevacizumab, azithromycin, and insulin, respectively [13,14,15].

Hydrogels (HGs) are polymeric materials capable of absorbing and retaining large amounts of water or other fluids [16]. In comparison to other materials, HGs are highly advantageous since they possess properties useful for biomedical applications, such as biocompatibility, biodegradability, stimulus responsiveness, and controlled release of active substances [17]. In addition, they are considered suitable materials for biomedical applications as they can successfully deliver therapeutic agents (e.g., plant extracts, isolated compounds, or peptides) experimentally or clinically [18,19] and pose ease of application through distinct administration routes and versatility to be formulated in a liquid or semi-solid form [20].

Currently, the global market size of HGs is estimated at USD 23.16 billion in 2024, but it is expected to increase to USD 32.62 billion by 2029. The evident economic growth of HGs in current markets can be attributed to their wide range of applications in the healthcare industry, especially for treating oral and maxillofacial diseases [21], periodontitis [22], recurrent aphthous ulcers [23], radiation or chemotherapy-induced oral mucositis [24], and chronic oral lesions of immunological origin [25]. As with other materials, the assembly, size, shape, and surface features of HGs can be controlled by optimizing their manufacturing, and there are various analyses (e.g., density functional theory and molecular dynamics analyses) designed and implemented to fulfill this purpose [26,27]. The factorial regression analysis is an advantageous statistical approach that enables the evaluation of the effect of multiple independent variables on a response variable, and it is highly suitable for identifying the predominant factors that can influence the performance and the mechanical, sensory, biocompatibility, and drug delivery properties of HGs [28].

HGs can be manufactured with natural (e.g., alginate, chitosan, or collagen) [29], or synthetic (e.g., polyethylene glycol, polyvinyl alcohol, or polyacrylic acid) materials [30]. Cationic guar gum (CGG), a derivative of guar gum, is a polysaccharide widely used for the formation of HGs due to its low cost and easy availability [31]. Structurally, CGG is constituted by a backbone of mannose units linked by *β*-1,4 glycosidic bonds with galactose units attached through *α*-1,5 glycosidic bonds. In contrast to its guar gum counterpart, CGG is characterized by its positive nature, which arises from the presence of quaternary ammonium groups and strongly contributes to its interaction with negatively charged molecules, such as mucin [32], and influences its capability to disrupt components from the cell membranes of pathogenic microorganisms, and wound healing activity [33]. In contrast to other cationic polymers, such as alginate and dextran, CGG is preferred due to its low toxicity, biodegradability, and controlled release properties. In addition, it is advantageous due to its versatility and modifiability for chemical modification and functionalization.

The complete formation of HGs for oral mucosa delivery requires the use of crosslinking agents such as polyethylene glycol (PEG). Contrarily to other hydrophilic polymers, PEG and its branched derivatives are synthetic polymers that possess significant biocompatibility, solubility, and hydrophilicity for materials design with applications in the clinical pipeline [34]. In addition, it is a suitable raw material that is characterized by its non-immunogenicity, versatility to be modified with distinct functional groups, and stability for long-term storage and use. For the formation of HGs, PEG is crosslinked with chemical or physical agents to aid the formation of the three-dimensional network of HGs. Propylene glycol (PG) is another polymer frequently used in combination with PEG, as it can drive ionic crosslinking reactions and result in stable HG networks. For the synthesis of drug delivery systems, the use of PG is convenient as it exhibits a high dissolving capacity of organic and inorganic analytes, prevents moisture, and executes non-toxic effects. However, additional weak organic acids such as citric acid (CA) can assist in such a reaction by creating ester linkages between PEG and PG and, hence, contribute to the stabilization of the HG structure and its mechanical features.

Toxicological analyses are required for identifying and characterizing the potential toxicity of substances and assessing dose-response relationships prior to their consideration in future pharmaceutical research. For nanobiotechnology purposes, in vitro and in vivo toxicological assays have been necessary to determine safe exposure limits and develop strategies for the safe therapeutic evaluation of silver, titanium, and zinc oxide nanoparticles [35,36,37]. In contrast to other in vivo models (e.g., Zebrafish and *Drosophila melanogaster*), *Artemia salina*, commonly known as brine shrimp, represents a major model to perform toxicity assays due to their ease of culture, sensitivity, and short life cycle [38]. In addition, it is preferred to evaluate the possible toxicity of bioactive materials such as drug delivery systems since it enables their large-scale screening through cost-effective, standardized, and simple experimental conditions.

The current study aims to optimize, develop, characterize, and evaluate HGs based on CGG/PEG for potential applications in oral mucosa. A 2^k^ factorial experimental design was implemented to optimize the viscosity, pH, and sensory properties of HGs. The chemical composition, mass loss, and surface features of HGs were investigated by a series of analytical techniques such as Fourier transform infrared spectroscopy (FTIR), Raman spectroscopy, thermogravimetric analysis (TGA), and scanning electron microscopy (SEM). The biological activities of HGs consisted of the in vitro evaluation of their mucoadhesive properties and antibacterial activity against Gram-positive (*S. aureus*) and Gram-negative (*E. coli* and *K. pneumoniae*) bacteria. The potential toxicity of HGs was investigated against *A. salina* nauplii as an in vivo model. Results indicated that the concentration of PEG influences the sensory features, structural arrangement, mass loss patterns, mucoadhesive capability, and antibacterial activity of the developed HGs. However, it does not influence their toxicity against *A. salina* nauplii. The results retrieved from this work provide novel insights into CGG/PEG-based HGs with desirable optimized features, mucoadhesive and antibacterial activities, and biocompatibility for oral mucosa drug delivery.

## 2. Results and Discussion

The oral mucosa can be affected by a variety of disorders such as viral (e.g., herpes, and hand, foot, and mouth disease) and bacterial (e.g., candidiasis, gingivostomatitis, and pericoronitis) infections [39,40], autoimmune diseases (e.g., lichen planus, pemphigus vulgaris, and Sjögren’s syndrome) [41,42], inflammatory conditions (e.g., stomatitis and lichenoid reactions) [43,44], and nutritional deficiencies (e.g., glossitis and cheilitis) [45,46]. In comparison to other materials, HGs possess suitable porosity, mechanical strength, tissue-like properties, and suitable biocompatibility for biomedical applications. Based on this, this work specifically aimed to optimize the physicochemical and sensory properties of the HG formulations by manipulating key variables, such as the concentration of CGG, PEG, and PG, within a 2^k^ factorial experimental design (see Table 1). The alterations in these parameters were hypothesized to influence the HG’s pH, viscosity, and sensory attributes, which are critical for its performance as an oral mucosal delivery system. Figure 1 depicts the formulation process of HGs.

### 2.1. Effect of Factors on Key Properties of HGs

Evaluating the effects of various factors on the response variables of HGs is crucial for optimizing their activity and further applications in the biomedical field. Key response variables that influence the performance of HGs include pH, viscosity, and sensory properties. For instance, the pH of HGs impacts their swelling behavior, mechanical strength, and drug-release capabilities [47], which has been observed for pH-sensitive PEG-based HGs designed to release drugs (e.g., doxorubicin, paclitaxel, and cisplatin) in response to specific pH conditions found in tumor microenvironments [48]. Comparably, viscosity is another important feature to investigate during the synthesis of HGs, as it can modify their elasticity, mechanical strength, capacity to modify cell adhesion and growth, and safety and effectiveness in clinical applications [47]. On the other hand, the evaluation of sensory properties (e.g., appearance, transparency, and color) is necessary since they dictate user experience, preference, aesthetics, and acceptance of the final product. In comparison to pH and viscosity evaluation, the sensory features of HGs are of great importance in skincare, cosmetics, and wound dressings [49].

Here, a factorial design analysis using Minitab^®^ assessed the impact of PEG type and ingredient concentrations on HG’s properties, including appearance, smoothness, and stickiness. In the same regard, sensory evaluation ratings were averaged and entered into Minitab^®^ along with pH and viscosity data. As illustrated in Figure 1A,B, factorial regression analysis revealed that pH was unaffected by any variables, whereas viscosity was significantly influenced by both PEG type and CGG concentration, with CGG concentration having the greatest impact (see Figure 1C,D). According to the Likert scale, the type of PEG is the main factor influencing appearance perception, followed by the content of CGG, PEG, and PG; see Figure 1E,F. In contrast, it was determined that the smoothness of HGs was directly related to CGG concentration. The significant influence of CGG concentration on viscosity, softness, and stickiness can be associated with its cationic characteristics. In the first case, coupled with its crosslinking caused by the addition of CA, an increase in viscosity was evident and, in the last two, are associated with the interaction of its charge with the negative charge of the application surface of the panelists, which promotes greater adhesion (stickiness) and subsequently, a smoothness associated with the possible formation of a film on the surface.

### 2.2. Optimization of the Formulation of HGs

Optimization in experimental design is crucial for ensuring efficiency, reliability, and obtention of meaningful data by focusing on relevant factors that ensure that response variables such as viscosity, pH, stickiness, smoothness, and appearance are optimized for desired outcomes [50]. Once the influence of the factors on the response variables was analyzed, the smoothness, stickiness, appearance, and viscosity of HGs were maximized (see Section 4.4). Based on the optimization processes, two formulations of HGs were chosen, and their composition is compiled in Table 2. According to Table 2, it can be noted that the HG with the lowest concentration of PEG-75 was designated as HG-1.0, whereas the one with the highest concentration was termed HG-2.5.

### 2.3. Evaluation of pH, Viscosity, Sensory, and Spreadability Analysis of Optimized HGs

The optimization of results in an experimental design is essential because it verifies if the experimental outcomes align with the predicted values, ensuring the reliability and accuracy of results. For instance, by re-evaluating viscosity, pH, stickiness, appearance, and smoothness, Minitab’s predictions can be confirmed, validating the experimental design and optimization process. Among the implemented analyses, this step is crucial for consistent and reproducible results, allowing for confident decision-making based on the data. As observed in Figure 2A,B, sensory tests, pH measurements, and viscosity assessments of the optimized HGs showed no significant differences (*p* > 0.05), indicating that an increase in the concentration of PEG-75 does not impact these variables. According to Figure 2C, HG-1.0 and HG-2.5 scored similar results during sensory evaluations, with appearance rated close to 5, smoothness around 4, and stickiness about 3. In view of these results, it is noteworthy to observe that the maximum achievable stickiness was relatively low but sufficient for oral mucosa applications due to CGG’s positive charge facilitating adhesion to mucin.

Spreadability is a crucial parameter in the evaluation of HGs, as it significantly influences both user experience and application efficiency. This characteristic determines how easily a hydrogel can be applied to a surface, which is essential for ensuring consistent and effective coverage in biomedical applications. In this study, spreadability tests indicated no significant differences (*p* > 0.05) between the HGs (see Figure 2D), implying that users would not perceive a difference during its application. These results can be attributed to the emollient properties of PEG-75, whose inclusion in HGs can enhance their moisturizing ability and texture, thereby ensuring consistent spreadability. Additionally, its compatibility with other HG components, such as CGG, supports formulation uniformity and facilitates ease of application. CGG improves hydrogel texture and spreadability through effective interactions with water and other ingredients. The combined effects of PEG-75 and CGG in maintaining viscosity and texture consistency likely contribute to the observed similarities in spreadability, consumer satisfaction, and HG acceptance.

### 2.4. Characterization of HGs

#### 2.4.1. SEM Analysis

In contrast to electron-based microscopy techniques, SEM provides essential insights into the surface morphology and internal structure of materials intended for biomedical applications. For the microscopy characterization of HGs, SEM analyses can aid in the determination of their pore size, distribution, and overall porosity [51]. As observed in Figure 3, SEM micrographs of lyophilized hydrogels revealed morphological differences linked to PEG-75 concentration. For example, it was noted that HG-1.0 had a smoother texture (see Figure 3A,B), whereas HG-2.5 exhibited a more porous structure (see Figure 3C,D). These variations suggest that PEG-75 concentration affects the internal structure, potentially influencing HG’s mechanical properties and future performance as carriers for pharmaceutical actives. This observation aligns with previous studies, which have demonstrated that the internal microarchitecture of hydrogels is significantly influenced by the concentration of PEG during synthesis.

#### 2.4.2. FTIR and Raman Spectroscopy Analysis

FTIR spectroscopy is based on the capacity of organic or inorganic molecules to absorb infrared radiation. According to the vibrational modes (stretch, bend, or scissoring) of analytes, FTIR analysis provides essential information about their molecular structure and chemical composition. In this study, FTIR spectroscopy was implemented to assess the chemical composition of HG-1.0 and HG-2.5, together with the raw materials utilized for their synthesis. As shown in Figure 4A, PEG-75 exhibits a characteristic band at 2884 cm^−1^, which can be associated with C-H stretching; the same peak is observed in both HGs without an evident shift or change. On the other hand, CGG presents a broad peak at 3272 cm^−1^ and another around 1632 cm^−1^, corresponding to vibrations of hydroxyl or amino groups, respectively. The FTIR spectra of HGs exhibit similar bands; however, variabilities in shape and size and shift to higher wave numbers compared to CGG suggest chemical interactions such as hydrogen and ionic bonding.

In accordance with the inelastic scattering of monochromatic light by organic and inorganic molecules, Raman spectroscopy can yield information about their crystallinity, phase transition, polymorphism, orientation, and possible defects or impurities. As represented in Figure 4B, the Raman spectra of HG-1.0 and HG-2.5 exhibit a small wide band around 2900 cm^−1^, which is characteristic of methylene group stretching. This signal is related to the PEG-75 signal (2888 cm^−1^); in this case, the intensity of this signal is higher for HG-2.5 compared to HG-1.0, which can be justified by the higher concentration of PEG-75 in the formulation. Additionally, both HGs show a small signal around 3060 cm^−1^ that is not present in any raw material and is possibly associated with O-H stretching, complementing the IR spectra observations regarding the hydrogen bonding interactions of the raw materials within the HG. Finally, the signals between 1570 and 1780 cm^−1^ for CA are detected in both HGs; however, they show changes in shape and intensity. These changes provide evidence of the interaction of CA, mainly with the cationic region of CGG, supporting the observations in the IR spectra.

#### 2.4.3. TGA and Mucoadhesive Analyses

TGA results revealed distinct mass loss patterns for the raw materials and optimized HGs (Figure 5A). Both CGG and the optimized HGs exhibited an initial mass loss of around 100 °C, attributable to the loss of non-bound water. CA showed a similar water loss slightly earlier. Both CGG and CA experienced total mass losses at approximately 200 °C and 300 °C, respectively, corresponding to their major decomposition points. For PG, a mass loss of 98% was observed around 180 °C, with the remaining 2% persisting at higher temperatures. PEG-75 exhibited a 10% mass loss at 400 °C, likely associated with the degradation of its carbon structure. 

The HGs displayed distinct mass loss curves at various temperatures, potentially due to the present significant mass losses at approximately 100 °C and 300 °C. However, at 400 °C, HG-2.5 experienced a complete mass loss of 100%, whereas HG-2.0 retained approximately 10% of its mass over the entire temperature range (see Figure 5A). These findings indicate that the thermal stability and degradation patterns of the HGs are significantly influenced by their composition, particularly the presence of PEG-75. The higher residual mass of HG-1.0 at 400 °C suggests a more thermally stable formulation compared to HG-2.5. This could be due to variations in the interaction between PEG-75 and other HG components, influencing the overall thermal degradation profile.

Mucoadhesion refers to the ability of a material to adhere to mucosal surfaces, like those in the gastrointestinal tract, nasal passages, vaginal cavity, or in this case, buccal cavity. HGs with strong mucoadhesive properties can improve drug delivery by prolonging the residence time of formulations at the target site; for example, studies have shown that in fluids (e.g., vaginal physiological fluids) that can wash away formulations, mucoadhesive agents enhance retention, leading to better treatment efficacy [52]. The improved therapeutic efficacy of HG adhered to mucosal surfaces is associated with the reduction in frequent dosing due to sustained drug release phenomena [53].

Here, the movement of HG-1.0 and HG-2.5 was analyzed by tracking the length of the drop, as it left a trail while moving, indicating good adherence to mucin. Over time, this trail descended slowly, accumulating more at the end of the drop, suggesting effective retention of the HGs on the mucin surface. The difference in PEG-75 concentration between HG-1.0 and HG-2.5 could influence their adhesion levels and reduced displacement on the mucin. As observed in Figure 5B, both HGs showed similar mucoadhesion kinetic profiles with a displacement factor at an equilibrium of 0.57 mm/mg; this reach of equilibrium occurred around 5 min. After this time, the displacement of the HGs remained constant or without significant changes. The observed phenomena can be attributed to the ionic interaction between the cationic group of CGG and the anionic moieties of mucin, resulting in mucoadhesion. In addition, these events can be associated with the presence of PEG-75, which can enhance the viscosity of HGs.

### 2.5. Antibacterial Activity

Infections caused by Gram-positive and Gram-negative bacteria constitute major economic and health challenges due to their capacity to develop a series of resistance mechanisms such as enzymatic inactivation, target modification, biofilm formation, and over-expression of efflux pumps to reduce the intracellular accumulation of antibiotics [54]. As illustrated in Figure 6A, treatment with 50 and 100 μg/mL of HG-1.0 caused the death of 16.32 ± 3.88 and 39.24 ± 3.17 *E. coli* cells. Comparably, treatment at 150 and 200 μg/mL resulted in the death of 41.93 ± 0.85 and 41.98 ± 0.95% cells, respectively. Contrarily to these results, HG-2.5 exerted enhanced antibacterial activity since treatment with 50, 100, and 150 μg/mL induced the death of 61.22 ± 1.49, 66.04 ± 2.02, and 69.20 ± 2.46% *E. coli* cells, respectively. The highest activity of HG-2.5 against *E. coli* was determined during treatment with 200 μg/mL, as it occurred in the death of 80.09 ± 5.65% of cells.

Towards Gram-positive bacteria such as *S. aureus*, treatment with 50 and 100 μg/mL of HG-1.0 resulted in the death of 4.89 ± 1.63 and 19.40 ± 13.63% cells, respectively. Similarly, treatment with 150 and 200 μg/mL of HG-1.0 promoted the death of 19.22 ± 8.94 and 20.12 ± 8.57% *S. aureus* cells. Contrarily to HG-1.0, the activity of HG-2.5 towards *S. aureus* was superior since treatment at 50, 100, and 150 μg/mL occurred in 77.81 ± 4.89, 79.93 ± 19.40, and 83.80 ± 20.12% death cells. The highest activity of HG-2.5 towards *S. aureus* was determined at 200 mg/mL, as it resulted in 95.16 ± 19.22% death cells (see Figure 6B). Against *K. pneumoniae*, treatment with 50, 100, and 150 μg/mL of HG-1 was ineffective as it did not decrease the viability of cells. However, treatment with 200 μg/mL of HG-1.0 occurred in 37.19 ± 0.06% *K. pneumoniae* cells. In comparison to the activity of HG-1.0, treatment with 50 and 100 μg/mL of HG-2.5 induced the death of 23.48 ± 0.03 and 31.03 ± 0.03% *K. pneumoniae* cells, respectively. At higher concentrations, treatment with 150 and 200 μg/mL of HG-2.5. occurred in 27.86 ± 0.02 and 45.19 ± 0.17% death cells (see Figure 6C).

In accordance with the Clinical and Laboratory Standards Institute (CLSI) guidelines, the sensibility of bacteria to treatment is classified based on the determined MIC values. In this regard, bacteria are categorized into susceptible (MIC ≤ 4 μg/mL), susceptible-dose dependent or intermediate (MIC 8–16 μg/mL), and resistant (MIC ≥ 32 μg/mL) [55]. Based on this information, MIC_90_ of HGs towards *E. coli*, *S. aureus*, and *K. pneumoniae* are compiled in Table 3 and suggest that the antibacterial activity of HG-1.0 and HG-2.0 is ineffective since the cultured bacteria are resistant to treatment. Despite this fact, the antibacterial activity of HG-1.0 and HG-2.5 against the cultured Gram-positive and Gram-negative strains can be associated predominantly with the content of CGG.

Even though CGG is widely used to manufacture complex nanomaterials for treating burn wounds or cancer [33,56], its antibacterial activity can be attributed to its polysaccharide nature. Recent studies have demonstrated that polysaccharides can disrupt the growth of pathogenic microorganisms through destabilization and disruption of the integrity of the cell membrane, causing enhanced permeability and leakage of cellular content [57,58]. In addition, it has been reported that polysaccharides are major secondary metabolites that can interfere with the ability of bacteria to attach to host cells or surfaces, which is of great importance in preventing the initial stages of infection [59].

The results recorded during this assay are of great importance since, in human health care, the incidence of Gram-negative strains such as *E. coli* is associated with severe cases of urinary tract infections (UTIs) [60], meningitis [61], or sepsis [62]. In comparison to other Gram-negative bacteria, *E. coli* is characterized by its wide presence in contaminated food, water, animal waste, and surfaces such as hands [63]. *K. pneumoniae*, another highly pathogenic Gram-negative bacterium, is responsible for causing (UTIs), sepsis, wound infections, and liver abscesses [64]. Compared to *E. coli*, *K. pneumoniae* is usually found in hospitals and long-term care facilities, in contaminated food and sources, and in gastrointestinal and respiratory tract systems [65]. In contrast to Gram-negative strains, Gram-positive bacteria such as *S. aureus* are frequently related to skin, soft tissues, bloodstream, surgical site, and food poisoning infections [66]. The variabilities in the activity of HG-1.0 and HG-2.5 can be attributed to the structural complexity of Gram-positive and Gram-negative bacteria as it has been documented that the former possess a simpler structure constituted by peptidoglycan layers, teichoic acids, and lack an outer membrane. In contrast, the latter exhibits a complex structure composed of an inner cytoplasmic membrane, outer membrane, and additional proteinaceous components such as porfirins that influence the passage of antibiotics.

### 2.6. Toxicity Evaluation In Vivo

The genus *Artemia*, also known as brine shrimp, encompasses a wide category of aquatic crustaceans distributed among hypersaline environments and characterized by their segmented body and crucial ecological role as food source for other aquatic organisms such as fish [67]. In contrast to other in vivo models, *A. salina* nauplii constitutes an attractive model for evaluating the potential toxicity of compounds or structures for biomedical applications. As depicted in Figure 7, treatment with HG-1.0 at the tested concentrations (50, 100, 150, and 200 μg/mL) did not cause any morphological aberrations among *A. salina* nauplii and resulted in a 100% survival rate. In the same regard, treatment with 50, 100, 150, and 200 μg/mL of HG-2.5 occurred in similar results. The obtained results suggest the biocompatibility of HG-1.0 and HG-2.5 within the range of 50–200 μg/mL.

Contrarily to other reports, the results obtained in this work are challenging to compare since CGG-based hydrogels are widely synthesized and evaluated for their drug delivery capacity, but there is no scientific evidence that validates their toxicity evaluation among in vivo models. However, when other raw materials have been considered to manufacture HGs, it has been unveiled that treatment with 1 and 5 μg/mL of nanocomposite hydrogel synthesized with laponite and polyethylene-glycol diacrylate loaded with irgacure, a frequently used phoinitiator, resulted in 80% live nauplii [68]. In the same context, treatment with hydrogels based on gelatin methacrylate and laponite within the range of 0.1–5.0 μg/mL has resulted in a 90% nauplii survival rate [69]. In another study, treatment with gelatin/carboxymethyl cellulose crosslinked with glutaraldehyde, resulted in a biocompatible effect as it did not affect *A. salina* survival rate [70]. The discrepancies with other studies can be attributed to the use of distinct tested concentrations, experimental conditions, and intrinsic features of raw materials utilized during synthesis.

## 3. Conclusions

This study developed and characterized HGs based on CGG/PEG for potential oral mucosa drug delivery applications. The factorial experimental design demonstrated that the concentration of CGG and PEG type significantly influenced key properties such as viscosity, appearance, smoothness, and stickiness of HG-1.0 and HG-2.5. Optimized formulations of HG-1.0 and HG-2.5 exhibited stable pH and viscosity, with no significant differences in sensory evaluation, suggesting their consistency in user perception and further applications. Regarding their characterization, SEM analyses revealed that HG-2.5’s higher PEG-75 concentration led to a more porous structure, suggesting slightly different mechanical properties and distinct drug delivery performance. In the same regard, TGA evaluation highlighted the impact of PEG-75 on the thermal stability of the HGs, with HG-2.5 showing distinct mass loss patterns, a phenomenon indicative of its suitability for different temperature conditions. FTIR and Raman analyses demonstrated the presence of characteristic peaks related to the raw materials used for the synthesis of HGs. When tested for their biological performance, mucoadhesive studies revealed that HG-2.5 presented enhanced retention on mucin surfaces, which is crucial for effective oral drug delivery and can be attributed to the ionic interaction between CGG with mucin together with the utilized PEG-75 concentration. The antibacterial property of the developed HGs was considered weak due to their MIC_90_ values (>500 μg/mL); however, HG-2.5 exerted the highest activity, especially against *S. aureus* followed by *E. coli* and *K. pneumoniae*. The recorded antibacterial activity of HG-2.5 suggests the need to continue evaluating its capacity to act as a drug delivery system to treat oral mucosa infections. On the other hand, toxicity evaluation using *A. salina* nauplii demonstrated the biocompatibility of HG-1.0 and HG-2.5 at tested concentrations (50–200 μg/mL) since treatment did not compromise their viability or their morphology. Taken together with these results, this study establishes for the first time the capacity of CGG/PEG-based HGs to act as effective, stable, and biocompatible vehicles for oral mucosa drug delivery. Given the determined size, porous features, thermal stability, retention behavior, and sensory properties, HG-1.0 and HG-2.5, can be suitable vehicles for the delivery of drugs. However, further investigations are necessitated to determine the long-term stability and therapeutic efficacy of these formulations and explore their effectiveness in delivering therapeutic agents such as antimicrobials, nanoparticles, or plant extracts.

## 4. Materials and Methods

### 4.1. Materials

CGG (N-Hance™ CG13, guar hydroxypropyltrimonium chloride, medium charge density, medium nitrogen content, and high molecular weight) was donated by Ashland^®^. PEG-75 lanolin (Solulan™ 75 lanolin) was obtained from Lubrizol™ (Ciudad de Mexico, Mexico). CA and PG-USP were from Reactivos Meyer^®^ (Ciudad de Mexico, Mexico) and AzuMex^®^ (Heroica Puebla de Zaragosa, Mexico), respectively. Porcine stomach mucin-type II was obtained from Sigma-Aldrich^®^ (St. Louis, MO, USA). All systems were prepared using water from a Milli-Q^®^ (MQ) filtration system (Millipore, Billerica, MA, USA).

### 4.2. Fabrication of the Hydrogel through Design of Experiments

A 2^k^ experimental design was performed by using Minitab^®^ (Version 17, Pennsylvania State University, University Park, PA, USA) software. Table 1 describes the factors and levels proposed for the study. As shown in Figure 1, HGs were prepared using a primary and a secondary container. In the primary container, CGG and PG were mixed with half the required water. In the secondary container, PEG-75 or PEG-150 (depending on the specific factorial design) was dissolved in the remaining water through heating and stirring. After equalizing the temperatures, the secondary container’s contents were poured into the primary container and mixed thoroughly. A 10% CA solution was then added dropwise to induce gelation, completing the hydrogel formation.

### 4.3. Viscosity, pH, and Sensory Analysis of HGs

Once the design matrix generated the HGs, they were manufactured, and the established response variables were evaluated for each. Viscosity was determined using a spindle viscometer (Byko^®^ Visc Basic, Geretsried, Germany) at 25 °C and 50 rpm. pH determination was conducted with a calibrated potentiometer (Ohaus^®^ Starter 2100, Parsippany, NJ, USA) at room temperature. For sensory analysis, three sensory properties (appearance, smoothness, and stickiness) were established. A group of 10 panelists evaluated these properties using a 5-point Likert scale, where 1 represented “very poor sensory” and 5 represented “very good sensory”. The results of the sensory study were statistically analyzed to determine the average opinion. The validation of the optimized levels, the viscosity, pH, and sensory analysis of the optimized HGs were evaluated using the same methodology.

### 4.4. Characterization of the Optimized Hydrogel

Following the optimization process using Minitab^®^ V-21.1.0 to assess the factors and select the best rheological, pH, and sensory properties for application on the oral mucosa, two HG formulations were chosen. The composition of each HG is detailed in Table 1. The key distinction between the two HGs lay in the concentration of PEG-75: 1% (HG-1.0) and 2.5% (HG-2.5). These HGs underwent characterization through various tests, including those conducted on non-lyophilized HGs and lyophilized HGs. For the lyophilization process, the HGs were frozen at −80 °C ± 1 °C and lyophilized using a Labconco™ FreeZone 6 L −84 °C lyophilizer (Kansas City, MO, USA), maintaining a collector temperature of −83.9 °C and a pressure of 0.075 mbar.

### 4.5. Spreadability

A spherical HG mass (0.12 ± 0.007 g) was positioned on a smooth surface, and another acrylic smooth disk of 7.59 g was placed atop. Subsequently, a range of forces (5.0–20.0 N) were applied at the center of the disk by using a flat probe adapted to a tensiometer ZP-500N (Shenzhen Ailigu Instrument Co., Ltd., Shenzhen, China). After applying force for 30 s, three diameter measurements were taken at various positions using a digital calibrator. For each force, the spreadability factor (*S_f_*, mm^2^/g) was determined using Equation (1). In this equation, *d* constitutes the average diameter, *W* represents the total weight (g), and *A* is related to the total area (mm^2^). All experiments were performed in triplicate.
(1)$$S_{f}=\frac{A}{W}=\frac{d^{2}\pi /4}{W},$$

Determination of the spreadability factor of HG-1.0 and HG-2.5.

### 4.6. FTIR and Raman Spectroscopy Analyses

Initially, the chemical composition of the developed HGs was determined using an Agilent Technologies Cary 630 FTIR (Santa Clara, CA, USA) spectrophotometer. Raman analyses were performed utilizing an XploRA™ Plus Horiba Raman Spectrometer (HORIBA Scientific, Piscataway, NJ, USA). For FTIR analysis, samples of raw materials, HG-1.0, and HG-2.5 were placed on a horizontal diamond to acquire spectra within the range of 500 to 4000 cm^−1^. The measurements represented an average of 150 scans with a resolution of 16. For Raman analysis, samples were placed on double-sided adhesive tape and attached to a microscope slide. The slide was positioned on the microscope stage and analyzed using a 100× LWD objective, 532 nm-Edge laser, within a range of 500–4000 cm^−1^.

### 4.7. TGA Analysis

HG samples were tested using a NETZSCH^®^ model STA 2500 Regulus themogravimeter (Netzsch Japan, Ltd., Yokohama, Japan). The samples were collocated in alumina crucibles. Measurements were carried out under nitrogen flow in the temperature range from 20 °C to 590 °C at a heating rate of 10 °C/min. Thermal analysis was also conducted for the raw materials.

### 4.8. SEM Analysis

The morphology of the HGs was characterized by scanning electron microscopy (SEM) using a Tescan^®^ MAIA3 (Brno-Kohoutovice, Czech Republic). Observations were made with an accelerating voltage of 6.5 kV after sputter coating with gold in a vacuum ion sputter coater (SEC, MCM-100). Specimens were mounted perpendicular to their surface on aluminum stubs and secured with carbon tape.

### 4.9. Mucoadhesive Properties Evaluation

The mucoadhesive properties of the HGs were determined through an in vitro study using mucin. In this assessment, a layer of 1.0% mucin solution was placed onto an acrylic slide (26.9 × 49 mm^2^), and subsequently, the plate was positioned at a 45° angle with a known mass of the HG deposited at the upper edge. The distance traveled by the HG over the mucin surface was measured using a digital calibrator and corrected with the mass of HG deposited. A kinetic mucoadhesion profile was generated from the acquired data. This procedure was performed in triplicate for both optimized HGs.

### 4.10. Analysis of Antibacterial Activity

The antibacterial activity of the developed HGs was determined against *Escherichia coli* (ATCC 25922), *Staphylococcus aureus* (ATCC 25923), and *Pseudomonas aeruginosa* (ATCC 14210) following the microdilution method. Briefly, bacteria were cultured in Mueller–Hinton broth (B&D) at 37 °C under orbital shaking and prepared to obtain a final optical density of 0.05 at 600 nm. In a 96-well plate, bacteria were dispensed in a 96-well plate with 50, 100, 150, and 200 μg/mL of HGs in a final volume of 100 mL of medium (per well). Treatment was maintained overnight, and the next day, a Multiskan Skyhigh microplate spectrophotometer (Thermo Fisher Scientific, Cleveland, OH, USA) was employed to determine absorbances at 600 nm. Fosfomycin, vancomycin, and amikacin were utilized as positive controls. All experiments were executed in triplicate.

### 4.11. Evaluation of Toxicity in A. salina

The possible toxicity of HGs was evaluated in *A. salina* nauplii, which was used as an in vivo model. Briefly, dried cysts from *A. salina* were obtained from a commercial supplier in Puebla, Mexico, and placed in 35 g artificial sea salt previously dissolved in 1 L of distiller water. The temperature was set at 32 °C, and cysts were maintained under vigorous aeration and illumination under cysts hatched. After this, 250 μL of nauplii were dispensed per well in a 96-well plate together with 50, 100, 150, and 200 μg/mL of the synthesized HGs. Changes in survival rate or aberrations in the morphology of *A. salina* were analyzed through a Leica DMi1 inverted microscope (Wetzlar, Germany) equipped with a FLEXACAM C1 camera. Leica software version 3.3.0 (Leica Microsystems, Wetzlar, Germany) was used to capture images.

### 4.12. Statistical Analysis

The Minitab^®^ V-21.1.0 computer program was used to perform statistical analysis. The influence of different factors on the behavior of HGs was performed by analysis of variance (ANOVA), while the difference between means was assessed using the Student’s *t*-test and confidence intervals at a 95% confidence level.

## Data Availability

The original contributions presented in the study are included in the article, further inquiries can be directed to the corresponding authors.

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
