# Peer review of "Development and Evaluation of the Biological Activities of a Plain Mucoadhesive Hydrogel as a Potential Vehicle for Oral Mucosal Drug Delivery"

_gels, 2024, doi:10.3390/gels10090574_

Round 1
Reviewer 1 Report
Comments and Suggestions for Authors
This research articles describes "Development and Evaluation of the Biological Activities of a Mucoadhesive Hydrogel as a Potential Vehicle for Oral Mucosal Drug Delivery "
This research addresses the development of semisynthetic mucoadhesive hydrogel. The authors should explain the following comments………
1- Is the prepared hydrogel is plain or medicated? The authors should explain this especially in the title of the article.
2- Can the prepared hydrogel be used as a vehicle for the delivery of drugs?
3- A paragraph about Artemia salina nauplii should be added in the introduction section.
4- Is the SEM imaging carried out on fresh or lyophilized samples?
5- The authors should explain why the prepared hydrogel gas a different antibacterial activities against different types of bacterial.
Comments on the Quality of English Languageminor editing of English language is required
Reviewer 2 Report
Comments and Suggestions for Authors
This article explores the development and optimization of hydrogels (HGs) for biomedical applications, focusing on their physicochemical and sensory properties. The research aimed to optimize key variables, including the concentration of components such as CGG, PEG, and PG, within a 2k factorial experimental design, hypothesizing that these factors would influence the HG’s pH, viscosity, and sensory attributes. The manuscript can be accepted after some minor changes.
Comments:
The topic is original and highly relevant to the field of drug delivery systems, particularly in oral mucosal applications. The research addresses a specific gap in the field by focusing on the development of HGs with optimized properties for mucoadhesion and stability, which are critical factors for effective oral mucosal drug delivery.
Compared to other published material, this study adds significant value by exploring the use of cationic guar gum in combination with PEG and other components to develop HGs. The methodology is robust, utilizing a 2k factorial experimental design, various spectroscopic techniques, SEM, and TGA for characterization. The conclusions are consistent with the evidence and arguments presented.
1. Provide more justification for choosing specific polymers (e.g., CGG, PEG, PG). Discuss why these materials were selected based on their known properties and how they contribute to the overall formulation.
2. If possible, include a flowchart or schematic representation of the formulation process. This visual aid can help readers better understand the steps involved in creating the HGs.
3. The tables are well-organized and present the data clearly. The figures effectively illustrate the results. Make sure that all figures and tables are accompanied by detailed captions that explain what is being shown, including any relevant conditions or parameters.
4. The references appear to be appropriate, covering relevant literature on mucoadhesive hydrogels, oral drug delivery systems, and the characterization techniques used in the study.
